# The Oxidative Stability of Champagne Base Wines Aged on Lees in Barrels: A 2-Year Study

**DOI:** 10.3390/antiox13030364

**Published:** 2024-03-18

**Authors:** Charlotte Maxe, Rémy Romanet, Michel Parisot, Régis D. Gougeon, Maria Nikolantonaki

**Affiliations:** 1Union Auboise Vignerons en Champagne, F-10110 Bar-Sur-Seine, France; charlotte.maxe@hotmail.fr (C.M.); michel.parisot@ua-champagne.fr (M.P.); 2UMR PAM 1517, Université Bourgogne Franche-Comté, Institut Agro, Université Bourgogne, INRAE, Institut Universitaire de la Vigne et du Vin–Jules Guyot, F-21000 Dijon, France; remy.romanet@sayens.fr (R.R.); regis.gougeon@u-bourgogne.fr (R.D.G.); 3DIVVA Platform, PAM UMR A 02.102, Institut Universitaire de la Vigne et du Vin-Jules Guyot, F-21000 Dijon, France

**Keywords:** Chardonnay Champagne base wines, oak barrel aging, on lees aging, white wine antioxidant metabolome, metabolomics

## Abstract

In contrast with the elaboration of still wines, the impact of barrel aging before the “prise de mousse” on the aging potential of Champagne base wines has not been studied so far. In the present study, the oxidative stability and related molecular fingerprints of Chardonnay Champagne base wines were reported after 1 year of on lees ageing in new oak barrels for two consecutive vintages. Regardless of the vintage, on lees ageing in new oak barrels improved the wines’ oxidative stability estimated by DPPH assay at 1 year, while UHPLC-Q-ToF-MS molecular profiling showed clear chemical modifications according to the ageing period. Oak wood molecular ellagitannins followed a linear extraction during barrel ageing for both vintages. However, the wines’ antioxidant metabolome composed by antiradical and nucleophilic compounds clearly appeared vintage- and barrel-aging dependent. These results enrich the understanding of white wines antioxidant metabolome and improve the knowledge of the ageing potential of Chardonnay Champagne base wines by integrating vintage- and barrel-ageing effects.

## 1. Introduction

Barrel aging on lees is acknowledged as a technique that contributes to the overall stability of premium wines, thanks to multiple mechanisms involving wood/wine and lees/wine interactions [1,2]. At the end of aging periods, which can last from a few months to a few years, wines have reached protein and tartaric stability, thanks to the enrichment in protecting colloids from lees [2].

Within this time frame, wines can also reach an oxidative stability, in particular white wines. This concept has been recently introduced in order to provide a rationale for the flow of chemical reactions involved throughout the processes of elaboration of wines, which are at the basis of its intrinsic capacity to retain its visual, olfactory and taste qualities during maturation and aging in the bottle [3,4]. The oxidative stability relies on a diversity of native compounds, originating from the grape but also from yeasts autolysis, which all contribute to the antioxidant metabolome of the wine [5]. From a physicochemical point of view, the oxidative stability can be addressed by the measure of antioxidant and/or antiradical properties of a wine, using a white-wine-optimized DPPH assay or EPR spectroscopy, respectively [6,7]. Both of these techniques measure instant indices of a wine’s ability to resist oxidation. In parallel and from a molecular point of view, the antioxidant metabolome can be characterized by a combination of mass spectrometry-based targeted and untargeted analyses aiming at the identification of antioxidant molecular markers having either radical scavenging or nucleophilic properties [3,6,8]. Recent studies have shown that a diversity of nitrogen- and sulfur-containing compounds are major contributors to this antioxidant metabolome of dry white wines [9,10]. In the particular case of on lees aging in barrels, the oxidative stability has been shown to be correlated to the wood tannin potential, for low and medium potentials [1]. Therefore, and although at low concentrations, extracted ellagitannins, which exhibit highly antioxidant properties, contribute either directly or through complex interplays to the stability of white wines [1,11]. However, our results also revealed that the ellagitannin extraction process is matrix dependent, with Sauvignon and Chardonnay peaking after six and eight months of barrel aging, respectively [1].

In this context, aging Champagne base wines on lees in barrels represents an enological practice, which still needs to be characterized for its impact on the oxidative stability of Champagne. If oak wood is common in Champagne, its use for the aging of fresh wines in new barrels is less widespread than in large-volume vats used for storing the “vins de réserve”. In Champagne, fresh wines, also named base wines, are produced by fermenting juices from grapes, which were harvested before complete ripeness [12]. Therefore, base wines exhibit specific properties compared to a still wine from the same grape variety, including, in particular, lower pHs and higher titratable acidity. Considering, for instance, the Chardonnay grape variety, a white wine from Bourgogne would often have pHs 0.2–0.4 higher than the corresponding base wine in Champagne, which suggests that the latter could exhibit distinct antioxidant metabolomes and related properties. To our knowledge, no study has addressed the antioxidant metabolome of Champagne base wines, and its evolution upon barrel aging on lees. Lower pH can indeed favor mechanisms such as hydrolysis of esters of glycosidic precursors [13]. Similarly, the reactivity of carbonyl species with nucleophiles such as phenolics can be favored through protonation of the carbonyl oxygen [13].

The aim of this work was to study CH-C-BW’s oxidative stability when fermented and aged in new oak barrels, over two successive vintages. For this purpose, targeted and untargeted approaches were applied. Targeted analyses were performed to determine CH-C-BW’s antioxidant capacities, and ellagitannins concentrations. Untargeted analyses were also run to screen CH-C-BW’s chemical fingerprints.

## 2. Materials and Methods

### 2.1. Wine Samples

CH-C-BW (Chardonnay Champagne base wines) were obtained for two consecutive vintages, 2020 and 2021. Whole grapes were pressed using a standard operating procedure consistent with Champagne appellation specifications. Musts were settled before their transportation from the pressing center to the winemaking center. Chaptalization was performed to achieve 11.0% *v*/*v* of potential alcohol (equivalent of total soluble content of 19.4 °Brix). Juices were barrel fermented after inoculation using a *Saccharomyces cerevisiae* selected dry strain (1–2 10^6^ cells/mL) (IOC 18-2007, Institut Œnologique de Champagne, Mardeuil, France and Zymaflore^®^ Spark, Laffort, Bordeaux, France). Before use, new barrels (Tonnellerie de Champagne) were first disgorged over 7 weeks by filling them with a Champagne reserve wine. Alcoholic fermentation (AF) was undertaken in a temperature- and humidity-controlled cellar (18 ± 1 °C, 80 ± 5%) and monitored daily. Diammonium phosphate (10 g/hL) (Institut Œnologique de Champagne, Mardeuil, France) was successively added when the voluminal mass reached 1050 and 1020 g/dm^3^. At the end of AF (9 and 7 days after the start, respectively, for vintages 2020 and 2021), total SO_2_ was measured for each barrel of wine, and adjustments were performed by adding sulfites to reach 60 mg/L of total SO_2_ in order to block the malolactic fermentation (MLF). In total, 12 and 10 barrels were analyzed for the 2020 and 2021 harvests, respectively, and all analyses were conducted at 3, 6 and 9 months of barrel ageing. The whole set of wine samples is summarized in Appendix A, and all oenological parameters are given in Appendix A (musts) and Appendix A (wines).

### 2.2. Chemicals

Castalin, castalagin, vescalagin and ellagic acid were purchased as all phyproof^®^ reference substances from PhytoLab GmbH&Co., KG (Vestenbergsgreuth, Germany). 2,2-diphenyl-1-picrylhydrazyl (DPPH) was furnished by Thermo Fischer Scientific (Ward Hill, MA, USA). Sodium phosphate dibasic, citric acid, 4-methylcatechol, potassium disulfite, Amberlyst^®^A-26(OH) and periodic acid were provided by Sigma-Aldrich (St. Louis, MO, USA). Absolute ethanol, formic acid, acetonitrile and 2-propanol were provided by Fischer Scientific (Waltham, MA, USA). A sodium hydroxide solution for HPCE (high-performance capillary electrophoresis) (1 mol/L, 1 N solution) from Agilent Technologies (Santa Clara, CA, USA) was used for the calibration of the mass spectrometer under a sodium formate solution. Carbon dioxide (CO_2_), nitrogen (N) and argon (Ar) were obtained from Air Liquide (Paris, France). Ultra-pure water (18.2 MΩ/cm) was obtained from a Milli-Q system (Merck, Darmstadt, Germany).

### 2.3. Wines Antioxidant Capacity: DPPH Assay

The measurement of wines’ antioxidant capacity by DPPH was performed according to the optimized method for white wines [9]. A solution of 250 mg/L of DPPH (concentrated DPPH solution) in methanol was prepared daily and agitated for 1 h. In parallel, wine samples were degassed to remove free SO_2_ using CO_2_ bubbling during a 10 min period (3 mL/min) [9,14]. Milli-Q water was deoxygenated using Ar bubbling during a 20 min period with a constant flow. A 1:10 dilution of concentrated DPPH solution was performed in a methanol citrate/phosphate buffer at 0.1 M and 0.2 M, respectively; a pH of 3.6 was used. Three calibration curves were created for each wine sample by adding increasing volumes of wines into 3.9 mL of DPPH solution. Milli-Q water was used for the preparation of blank samples. Samples were prepared in a glove box under nitrogen atmosphere. Samples were incubated for 4 h in the dark before being analyzed at 525 nm using a UV 1800 spectrophotometer (Shimadzu™, Kyoto, Japan). Ec_20_ which corresponded to the volume of wine that decreased the initial absorbance of DPPH by 20% was determined for each sample from the measured absorbance. Details of calculations have been described in a previous study [9]. Raw data are summarized in Appendix A.

### 2.4. Wine Phenolic Contents: Determinations of I_280_ and A_420_

To quickly determine the wine phenolic content, the total polyphenol index (I_280_) was used. It was obtained from the direct measurement of wine absorbance at 280 nm [15] using a UV 1800 spectrophotometer (Shimadzu™, Kyoto, Japan). This index was measured only on wines from vintage 2021. Samples were 1:10 diluted in distilled water. The index corresponded to the measured absorbance multiplied by the dilution factor. The determination of the absorbance at 420 nm (A_420_) of wines from vintage 2021 was also realized. This measurement is considered to be directly linked both to quinones, which are products of oxidation of phenolic compounds, and to the “yellow” color characteristic of wines [16,17]. Wine oxidation state and yellow color were determined by measuring the absorbance at 420 nm in a 10 mm-path-length cuvette against the blank (distilled water), using a UV 1800 spectrophotometer (Shimadzu™, Kyoto, Japan) [18].

### 2.5. Quantification of Molecular Ellagitannins by UHPLC-Q-ToF-MS

Ellagitannins quantification was realized using ultra-high-pressure liquid chromatography (Dionex Ultimate 3000, Thermo Fischer Scientific, Waltham, MA, USA) coupled to a MaXis plus MQ ESI-Q-TOF mass spectrometer (Bruker Daltonics GmHb & Co, Bremen, Germany). Before analysis, wine samples were centrifugated at 6000× *g* for 10 min. The column used was a Restek^TM^ Pinnacle^TM^ DB Biphenyl 1.9 µm, 100 × 2.1 mm (Restek, Bellefonte, PA, USA) in reverse phase to analyze polar compounds. The mobile phase was (A) acidified water (0.1% (*v*/*v*) formic acid) and (B) acidified methanol (0.1% (*v*/*v*) formic acid). Separation was carried out at 40 °C with an initial flow of 0.3 mL/min following the gradient: 0–7.0 min of 4% (*v*/*v*) of eluant B; 7.0–9.0 min 7% (*v*/*v*) of eluant B with a flow of 0.2 mL/min; 9.0–12.0 min 50% (*v*/*v*) of eluant B with a flow of 0.3 mL/min and finally 12.0–13.0 min from 50% (*v*/*v*) of eluant B to 4% (*v*/*v*) of eluant B with a flow of 0.3 mL/min. Targeted analysis of ellagitannins was performed with a MaXis plus MQ ESI-Q-ToF mass spectrometer (MS) (Bruker, Bremen, Germany) equipped with an electrospray ionization (ESI) source (3.4 bars pressure for nebulizer and 10 L/min for nitrogen dry gas flow) and operated in negative ion mode (dry gas temperature = 200 °C, capillary voltage = 3500 V, end plate off set = 500 V) using both full scan (200 and 1500 *m*/*z*) and MS (collision energy: 8.0 eV) modes. Before batch analysis, the mass spectrometer was calibrated using sodium formate in enhanced quadratic mode. Quality controls were analyzed before and throughout each batch (every 10 samples), to verify the stability of the UHPLC-MS-Q-ToF system. Samples were analyzed randomly.

Quantitative analyses were performed using extracted ion chromatograms with a mass tolerance of 5 ppm, *m*/*z* 300.9992 (RT 10.6 min) for ellagic acid, *m*/*z* 631.0589 (RT 1.4 min) and *m*/*z* 631.0594 (RT 1.7 min) for castalin, *m*/*z* 933.5659 (RT 7.3 min) and *m*/*z* 933.0639 (RT 7.3 min) for castalagin, *m*/*z* 466.0291 (RT 4.5 min) and *m*/*z* 933.5660 (RT 4.5 min) for vescalagin). Calibration curves were generated using standard solutions containing 0.01 mg/L–15 mg/L for each compound diluted in model wine (12% *v*/*v* ethanol, 0.01% formic acid, pH adjusted to 3.2). Calibration curves were constructed by linear regression analysis of the peak area of each analyte. Equations of calibration curves determined for the quantification of each compound were determined by batch. The limit of detection (LOD) and limit of quantification (LOQ) of each compound were calculated by the linear regression method [19]. The LOD was estimated at 1.198 mg/L for ellagic acid, 0.414 mg/L for castalin, 0.317 mg/L for castalagin and 0.293 mg/L for vescalagin. The LOQ was estimated at 3.995 mg/L for ellagic acid, 1.380 mg/L for castalin, 1.058 mg/L for castalagin and 0.978 mg/L for vescalagin. Raw data are summarized in Appendix A.

### 2.6. Untargeted Analyses of Wines by UHPLC-Q-ToF-MS

Wines samples were analyzed by ultra-high-pressure liquid chromatography (UHPLC) (Dionex Ultimate 3000, ThermoFisher, Waltham, MA, USA) coupled to a MaXis plus MQ ESI-Q-ToF mass spectrometer (MS) (Bruker, Bremen, Germany). Before analysis, samples were centrifuged at 6000× *g* for 10 min. Reverse-phase liquid chromatography, UPLC Acquity BEH C18 1.7 µm column 100 × 2.1 mm (Waters, Guyancourt, France), was used to separated non-polar compounds. Mobile phase was (A) acidified (0.1% (*v*/*v*) of formic acid) water/acetonitrile (95/5 *v*/*v*) and (B) acidified (0.1% (*v*/*v*) of formic acid) acetonitrile. The flow rate was 0.4 mL/min and the temperature of elution was 40 °C. The column gradient was as follows: 0–1.10 min 5% (*v*/*v*) of eluent B and 95% of eluent B at 6.40 min with the end at 10 min. The dry nitrogen flow was 10 L/min and the nebulizer pressure at 2 bars. Ionization was performed in negative and positive modes. Parameters of ions transfer were as follows: 500 V as endplate offset and capillary voltage at 4500 V (positive ionization mode) and 3500 V (negative ionization mode). Acquisitions were realized between 100 and 1000 *m*/*z*, using 8 Hz spectra rate with auto MS/MS function (20–50 eV). The mass spectrometer was calibrated before each batch analysis using Na formate clusters solution (errors < 0.5 ppm). Analysis of Na formate clusters solution, at the beginning of each run, allowed recalibration of the spectra. The check of the stability of the UHPLC-Q-ToF-MS system was performed by quality controls (mix of all samples) which were analyzed before, after and every 10 samples during batch analysis. All the samples were frozen for storage before analyses and were analyzed randomly in one batch.

Bruker Compass MetaboScape software (v. 8.0.1, Bruker, Mannheim, Germany) was used for data pre-treatment, including mass spectrum recalibration, extraction of features (couple mass-to-charge ratio (*m*/*z*) and the retention time (RT)) and annotation using SmartFormula (based on isotopic profile), spectral library (MassBank) and a homemade database. The parameters of annotation were <5 ppm for *m*/*z* tolerance, <20 for mSigma, <0.3 min for retention time and >600 for MS/MS score. The features extracted corresponded to those with an intensity higher than 1000 which were present in at least 20% of total samples, and 40% for recursive parameters. Features were filtrated to keep only the features with intensity more than 3 time higher in samples compared to blanks (S/N>3). A Matlab script (version R2021a, MathWorks, Natick, MA, USA) was used to annotate features using KEGG and a home-made peptides database. Five different identification confidence levels were used for high-resolution mass spectrometric analysis in this paper according to Schymanski et al.’s 2014 publication [20]. Level 5 referred to an exact mass. Level 4 was attributed when an unequivocal molecular formula was found. Level 3 referred to an annotation thanks to a database. Level 2 corresponded to a database annotation confirmed by a spectral library match. Finally, level 1 was the identification of the compounds thanks to a reference standard [20]. These identification confidence levels will be mentioned for each feature discussed and specified after its measured mass as level 1, 2, 3, 4 and 5 [20,21]. All features involved in the impact of barrel ageing, which will be discussed in the results, are synthetized into Appendix A, with annotations’ levels referred to Schymanski et al., 2014 [20].

### 2.7. Antioxidant Metabolome

Untargeted analyses of wines and the characterization of their antioxidant metabolome were realized following the protocol proposed by Romanet et al., 2023 [5]. The antioxidant fraction associated with antiradical scavenging properties (AM-RS) was determined from features found in the 66 CH-C-BWs that were identified as both radical scavengers and showing increasing trend with the barrel ageing. The second antioxidant fraction, based on nucleophilic properties (AM-NU) was analyzed using the protocol proposed by Romanet et al., 2020 [8]. Derivatization of wine samples (without pH adjustment) was realized using 4-methyl-benzoquinone (4Me-Q) prepared in acetonitrile as proposed by Nikolantonaki and Waterhouse, 2012 [22]. To 1 mL of wine, 1 mM of 4Me-Q was added and incubated in the dark during a 30 min period. The reaction was then quenched by SO_2_ addition (1 mM). The derivatized samples were analyzed by an UHPLC-Q-TOF-MS system within 24 h, as described in Section 2.6. Wine samples were analyzed freshly (without freezing) with the same protocol as previously described.

After pre-treatment of the data using MetaboScape, AM-NU were extracted using a Matlab script by comparing derivatized and underivatized samples, by partial least squares discriminant analyses (PLS-DA) and Wilcoxon test using the following parameters: VIP > 1 and *p*-value < 0.05. Masses of AM-NU before derivatization were calculated (mass detected − mass of 4MeQ) and were annotated using KEGG, Phenol-Explorer and a home-made peptides database. The identification confidence levels were mentioned for each AM-NU feature as described in the previous paragraph and were written after its measured mass as level 1, 2 or 3. All discussed AM-RS and AM-NU are summarized in Appendix A with their characteristics. In this table, levels of confidence are adapted from Schymanski et al., 2014 and Sumner et al., 2014 [20,21].

### 2.8. Statistical Analyses

The results of wine antioxidant capacity by DPPH assay, total phenolic content and absorbance at 420 nm were expressed as mean ± standard deviation. For each duration of barrel ageing, n, the number of samples was equal to 12 for vintage 2020 and 10 for new barrels for vintage 2021. Comparisons between means for the DPPH assay, total phenolic index and absorbance at 420 nm were performed with a non-parametric test Kruskal–Wallis, followed by a *post-hoc* Dunn test, by adjusting the *p*-value with the Bonferroni method with RStudio scripts. ANOVA was performed on untargeted UHPLC-Q-ToF-MS data followed by a principal component analysis (PCA) to visualize samples using a homemade RStudio script. For all metabolomics statistical analyses, a feature was considered as “present” in a modality when its mean intensity measured by UHPLC-Q-ToF-MS/MS was higher than zero. From these results, features were extracted and plotted as a heatmap. To override the vintage effect, each dataset of 2020 and 2021 vintages was standardized by Pareto normalization. The two data subsets were compiled afterwards.

## 3. Results and Discussion

### 3.1. Wine Antioxidant Capacity of New Oak Barrel Ageing

The antioxidant capacity evaluated by the Ec_20_ parameter was determined for each CH-C-BW at 3, 6 and 9 months of ageing in new oak barrels for vintages 2020 and 2021 (Figure 1).

At the very beginning of barrel ageing (3 months) in new oak barrels, the average Ec_20_ values in the CH-C-BWs were 22 ± 1 and 16 ± 1 for 2020 and 2021 vintages, respectively (Appendix A). These results indicate that the wines from vintage 2021 had a better initial antioxidant capacity than those from vintage 2020. Considering grape musts for both vintages come from the same field, the major difference between the native antioxidant capacity after 3 months of barrel ageing could be explained by the vintage. It is known that climate (temperature and rainfall) and vintage are one of the major parameters to impact wine antioxidant capacity [23]. With respect to the classification model established on dry Chardonnay wines from Burgundy by Romanet et al., 2023 [5], wines from the two vintages presented high antioxidant capacities from the early stages of barrel aging. The release of antioxidant peptides during autolysis, along with the extraction of oak wood ellagitannins could certainly contribute to this behavior [5,24,25,26,27]. However, as shown by I_280_ (total phenolic index) and A_420_ measurements for the 2021 vintage (Appendix A) the polyphenolic content of these wines cannot explain such antioxidant capacity [5].

In order to better understand the role of ellagitannins on CH-C-BW’s oxidative stability, the kinetic profile of three molecular ellagitannins (vescalagin, castalagin and castalin) were monitored during barrel ageing. Figure 2 shows a significant linear and continuous increase in the sum of the three analyzed ellagitannins during 9 months of barrel ageing. The kinetic profile of total molecular ellagitannin contents was identical for both vintages and in accordance with already reported values in the case of still Chardonnay wines [1]. Considering the important number of replicates, n = 12 and n = 10 for 2020 and 2021 vintages, respectively, and the absence of differences on the sum of molecular ellagitannin concentrations for both vintages, we assumed that all the tested new oak barrels were characterized by similar tannin potentials. The sum of the three ellagitannins concentrations expressed as milligrams per liter of ellagic acid in wine revealed increasing concentrations ranging from 9.1 mg/L (vintage 2020, 3 months) to 19.3 mg/L (vintage 2021, 9 months), similar to those reported in still white wines aged in new French barrels [1,28,29]. For both vintages and in accordance with the literature, vescalagin and castalagin were the most concentrated [30,31]. Barrel ageing induces an extraction of each ellagitannin, and during its course, castalagin is hydrolyzed into castalin and ellagic acid, which explains the increase in the former with the contact duration [32]. According to previous results [33], ellagitannins and especially vescalagin and castalagin concentrations should be correlated with wines’ antioxidant capacity; our results showed that this was likely the case for the 2020 vintage, whereas the increase in ellagitannins concentrations for the 2021 vintage did not contribute, or to a lesser extent, to an increase in the antioxidant capacity (Figure 1 and Figure 2).

### 3.2. UHPLC-Q-ToF-MS Untargeted Analysis of Chardonnay Base Wines during Barrel Ageing

Analyses of the 66 CH-C-BW by UHPLC-Q-ToF-MS allowed us to have an overview of the CH-C-BW metabolome and to isolate features correlated to the barrel ageing. Unsupervised principal component analysis (PCA) of the 66 CH-C-BW’s MS data comprising up to 7269 features, is plotted in Figure 3. The first dimension (Dim1) of the PCA, which explained 24.6% of all the samples’ variance, allowed clear discrimination of the vintage, while the second dimension (Dim2) of the PCA, which explained 7.8% of the samples’ variance indicated a clear effect of barrel ageing on the wines’ chemical composition. This first study of the metabolome of champagne base wines aged in new oak barrels reflected an important evolution of their chemical composition throughout the 9 months of barrel ageing, which further appeared consistent over two successive vintages. It is worth mentioning, that the chemical composition of the wines from vintage 2020 was clearly impacted by the unwanted malolactic fermentation that took place between 3 months and 6 months, whereas no malolactic fermentation occurred in 2021. Therefore, Figure 3 highlights that during barrel aging, the malolactic fermentation can have a hierarchically more important impact on the chemical fingerprint of a wine than a wood wine contact duration of 3 months, as shown by the absence of discrimination between the analyses at 6 and 9 months for vintage 2020.

Features were filtrated to keep only those impacted by the barrel ageing (ANOVA, *p*-value < 0.01), allowing us to isolate 1391 significant ones. The heatmap representation of the dynamic behavior of these 1391 significant features allowed the discrimination of wine samples according to the time of barrel ageing and the vintage (Figure 4a). During on lees ageing in oak barrels, we observed a significant modification of the chemical fingerprints of wines, in agreement with the study published by Romanet et al., 2023 [5]. We observed a group of 1007 features exhibiting an increasing trend with ageing (Figure 4b).

These features were characterized by low apparent masses (89% of features had a detected mass under 400 Da) with retention times centered around 1 and 3 min, thus associated with rather polar compounds (Figure 4b).

Quercetin, salicylic acid and gentisic acid were identified in this group of features. It has been proven that the increase in quercetin contents could be due to the flavonol glycosides being hydrolyzed into flavonol aglycones during the barrel ageing [34]. Furthermore, a low pH enhances the rate of flavonoid glycoside hydrolysis to their corresponding aglycones [35,36,37]. Salicylic acid is one of the oak wood components and is partly transformed by sugar degradation during barrel toasting [38,39], but can be also be found at its native form in white wines [40]. Gentisic acid is one of the phenolic acids contained into oak wood that can be extracted during barrel ageing and has been already found in white wines [41,42]. It is worth mentioning that ellagic acid and gallic acid were also identified as significant increasing compounds during the barrel ageing, consistently with the hydrolysis of vescalagin and castalagin occurring during aging in new oak barrels [43]. An additional mass was annotated with the C_41_H_26_O_26_ elemental formula, which could correspond to castalagin and/or vescalagin, the two mains isomeric ellagitannins found in oak wood-aged wines. Another identified feature which corresponds to shikimic acid presented a high interest in this study, since it is considered to play a key role in wines. Shikimic acid (4,5-trihydroxy-1-cyclohexene-1-carboxylic acid) was identified as an oak barrel-ageing marker in our study and has been also presented as a biomarker of Chardonnay wines [44]. As a consequence of yeast autolysis, occurring between 3 and 9 months, proteins are hydrolyzed into peptides [45,46,47,48,49,50,51]. This fact is in accordance with the 191 features found in CH-C-BWs that have been annotated by our home-made peptides database, which include, alanine (A), aspartic acid (D), cysteine (C), glutamic acid (E), glutamine (G), histidine (H), leucine (L), lysine (K), phenylalanine (F), tryptophan (W) and tyrosine (Y). After annotation, the C_12_H_24_O_3_N_2_ elemental formula possibly corresponding to the Leu-Leu (LL) peptide exhibited an increasing trend during barrel ageing. Leucine is one of the major amino acids found in autolysates in model wine conditions, with alanine and γ-aminobutyric acid [50,52,53,54]. For both vintages, peptides with a mass range between 200 and 650 Da represented 19% of features following an increasing trend with the barrel ageing. In summary, this group of “increasing features” appeared to be composed of vitamins, small peptides and phenolic compounds such as ellagic and gallic acids, all exhibiting antioxidant properties, likely contributing to an improvement of the CH-C-BW’s antioxidant capacity during on lees barrel ageing.

On the other hand, 384 features presented a decreasing profile with barrel aging, as illustrated in Figure 4c. Overall, 37% of decreasing features included compounds with *m*/*z* higher than 400 Da in contrast to only 12% of the increasing compounds. As wines from 2020 underwent MLF, two features annotated as malic acid and citric acid were consistently part of this group. In the class of phenolic compounds, one feature identified as caftaric acid also followed a decreasing trend with the barrel ageing, in accordance with the literature [55,56]. Indeed, this tartaric acid ester is a very reactive compound and participates in oxidation processes [55]. Moreover, it is important to notice that vintage 2021 was significantly more acidic than vintage 2020. This fact can be explained firstly by the climatic conditions of the vintage and also the wines from vintage 2020 underwent MLF. The acidic conditions are conducive to faster hydrolysis kinetics. In total, 81 features with a decreasing trend were also annotated as peptides, thus contributing to 21% of the pool of features that presented a decreasing profile. These results emphasize the complex chemistry associated with the evolution of the concentration of peptides during barrel aging, with increasing and/or decreasing trends, depending in particular on their molecular weights and chemical natures.

### 3.3. Chardonnay Base Wines Antioxidant Metabolome during Barrel Ageing

Champagne base wine antioxidant metabolome was characterized according to the methodology proposed by Romanet et al., 2023 [5]. Indeed, champagne base wine antioxidant metabolome should be the sum of molecular antioxidant markers characterized by their radical scavenging (AM-RS) and nucleophilic (AM-NU) properties. Different compounds already mentioned above, with increasing concentration with barrel ageing, are known for their antiradical scavenging properties (AM-RS). They include gentisic acid, gallic acid and quercetin, which have a strong antiradical activity whereas salicylic acid has a weaker one [57,58]; vescalagin or castalagin [11,59,60], along with their hydrolysis product ellagic acid, known to be a good radical scavenger even at low concentration (1 µM) [59,61,62]; and all features annotated as peptides and amino-acid derivatives such as N-acetyl-L-methionine [63]. The feature annotated as C_5_H_4_O_3_ is particularly interesting. It could correspond to the 2-furancarboxylic acid, a compound derived from the oxidation of 5-hydroxymethylfurfural (5-HMF), already found in white wines [64,65,66]. 5-HMF is a well-known intermediate in the Maillard reaction and has been already found in sparkling wines [67,68,69] or could come from the oak as a thermal degradation product of furanic compounds during barrel toasting [70,71].

The targeted analysis of AM-NU allowed us to isolate 235 AM-NU molecular features, presented in Appendix A. From these 235 nucleophilic features, 19 were already identified in white wines [8]. Annotation using online databases allowed us to annotate 32 nucleophilic compounds, containing peptides or phenolic compounds. The number of AM-NU detected and the sum of their relative intensities were not impacted by the vintage and the aging time (Appendix A). However, a PCA based on the 235 isolated features clearly revealed an impact of the vintage (PC1), and an impact of the aging duration (PC2) (Figure 5a). To determine the evolution of each AM-NU independently to the vintage, Pareto normalization was realized. For this, the dataset was split into vintage 2020 samples and vintage 2021 samples, and the intensity of each AM-NU was normalized by Pareto scaling in each sub-dataset, before recombining the dataset. PCA based on this normalized dataset allowed us to visualize the impact of aging without vintage effect, showing a clear impact of the aging duration (Figure 5b). Based on these normalized data, it was possible to determine how the AM-NU were affected by the aging duration, by calculating the difference in the normalized intensity between T3, T6 and T9 (Figure 6). The AM-NU with a positive value for T3-T6 and T6-T9 were present in higher concentrations at the end of barrel aging (77 AM-NU colored in green) while a negative value indicated a lower concentration at the end of barrel aging (50 AM-NU colored in red).

Some peptides were found among the AM-NU consumed during barrel aging, thus considered to be involved in the resistance of the wine against oxidation. They included sulphur-containing compounds, such as an amino acid which could be potentially annotated as L-cysteine sulfinic acid, a key intermediate in the biosynthesis of cysteine and already found to participate in the AM-NU fraction of white wines [3,72], and glutathione or a peptide which could correspond to CNS or CGGS, that could come from yeasts autolysis [26,52,73,74].

AM-NU markers, which were more present at T9 compared to T3, should correspond to lees-derived or wood-extracted compounds, although they can be also involved in the resistance of the wine against oxidation. They consistently include some putative peptides like FPV or AFP, phenolic compounds such as gentisic acid (or isomers) and gentisyl alcohol (or isomers). Enrichment into antioxidant metabolites during barrel ageing and on lees provided a better capacity for ageing, as it is very important for Champagnes due to their process and especially on laths ageing that can last at least 15 months to many years [75]. Moreover, the sensory perceptions of base wines by oak-wood extracted compounds, such as lignans and coumarins, could also be affected such as sweetness, bitterness and acidity; these compounds are known for their antioxidant properties [76,77,78,79,80,81,82,83,84].

These results are in accordance with those recently described by Romanet et al., 2023 [5], indicating that the population of wine-relevant nucleophiles is modulated during barrel ageing according to the chemical environment and a part of that is due to the chemical mechanisms related to wines oxidative stability by the formation of reactional products with great antioxidant capacity or by neutralizing others (i.e., oxidized polyphenols). The analysis of the CH-C-BW’s antioxidant metabolome partly shares the Chardonnay wines from Burgundy’s antioxidant metabolome as previously studied and detailed by our team [3,5,6,8].

## 4. Conclusions

Champagne base wines from vintage 2020 and 2021 were screened with targeted and untargeted analyses. The DPPH assay showed that ageing Chardonnay Champagne base wines in new oak barrels improved or maintained their antioxidant stability. For both vintages, champagne base wines exhibited an antioxidant activity similar to Chardonnay wines from Burgundy. For both vintages, the extraction of total ellagitannins showed a linear profile. At the end of barrel ageing, total ellagitannins concentrations of Chardonnay Champagne base wines for vintages 2020 and 2021 were, respectively, 18.8 ± 5.2 mg ellagic acid equivalents/L and 18.3 ± 7.4 mg ellagic acid equivalent/L. Untargeted metabolomics analysis on 66 CH-C-BWs showed a chemical change during the barrel ageing with 1007 features that significantly increased with the barrel ageing and 384 features that significantly decreased with the barrel ageing. Annotations suggested a high contribution of peptides, known to be released by the on lees ageing process, phenolic compounds and probably oak wood compounds such as castalagin and/or vescalagin, all known to be antioxidant compounds. In total, 235 features were found to contribute to the molecular antioxidant markers with nucleophilic properties (AM-NU). Globally, the population of the AM-NU molecular fraction remained stable during barrel ageing, however, individual nucleophiles can follow stable, increasing or decreasing trends according to the chemical environment related to the wine matrix complexity and the barrel ageing conditions.

## Figures and Tables

**Figure 1 antioxidants-13-00364-f001:**
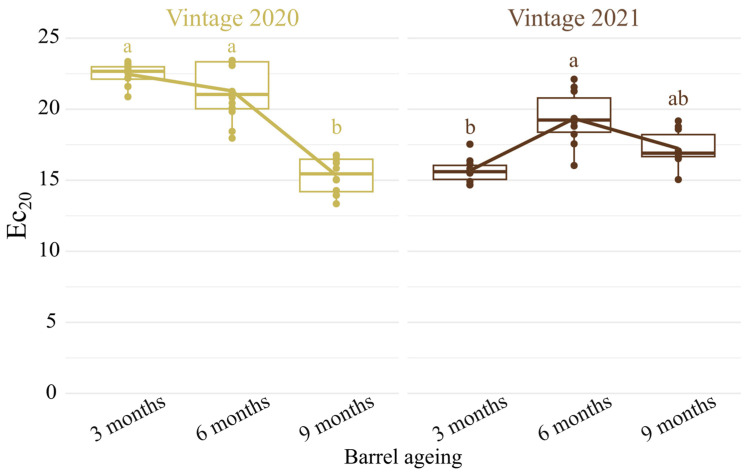
Evolution of the Ec_20_ parameter of Chardonnay base wines from vintage 2020 (n = 12) and vintage 2021 (n = 10), aged in new oak barrels for 9 months. Given that the AF lasted 9 and 7 days, respectively, for vintages 2020 and 2021, analyses at 3 months were considered to represent the beginning of barrel aging. Different letters indicate significant differences within a given vintage (Kruskal–Wallis followed by a *post-hoc* Dunn test, *p*-value < 0.05). Mean and quartiles constitute the boxplots and points represent measured values.

**Figure 2 antioxidants-13-00364-f002:**
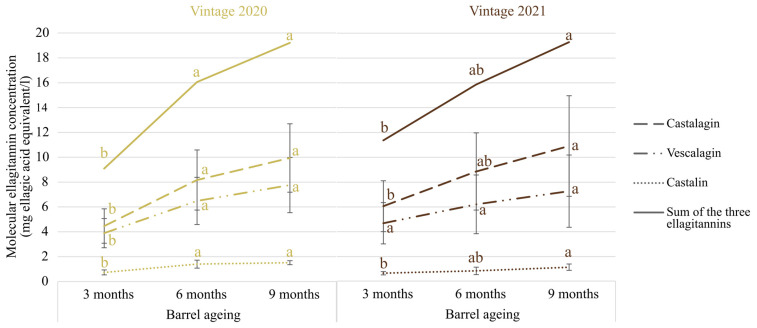
Evolution of the concentration of molecular ellagitannins (mg ellagic acid equivalents/L) in Chardonnay Champagne base wines (CH-C-BWs) from vintage 2020 (n = 12) and vintage 2021 (n = 10) aged in new oak barrels over 9 months. Kinetics of each ellagitannin were considered by vintage and significant differences were evaluated by a Kruskal–Wallis test (*p*-value < 0.05). Significant differences were determined by a unilateral Mann–Whitney U test (*p*-value < 0.05) comparing the sum of the three ellagitannins within a given vintage. Standard deviations of the two curves of sum of the three ellagitannins are not present to avoid overlapping. Different letters indicate significantly different groups determined using Kruskal-Wallis or Mann-Whitney test (*p*-value < 0.05).

**Figure 3 antioxidants-13-00364-f003:**
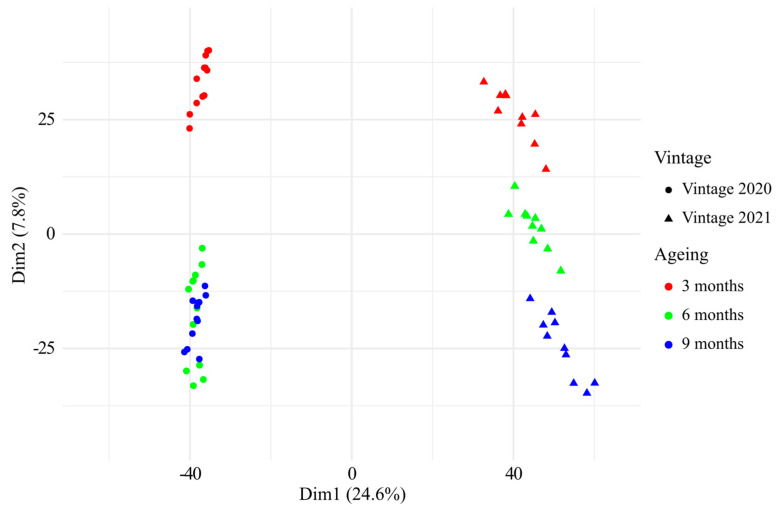
Principal component analysis of the UHPLC-Q-ToF-MS data of all the 66 CH-C-BWs from vintages 2020 and 2021 aged in new oak barrels for 3, 6 and 9 months, based on intensities of 7269 features.

**Figure 4 antioxidants-13-00364-f004:**
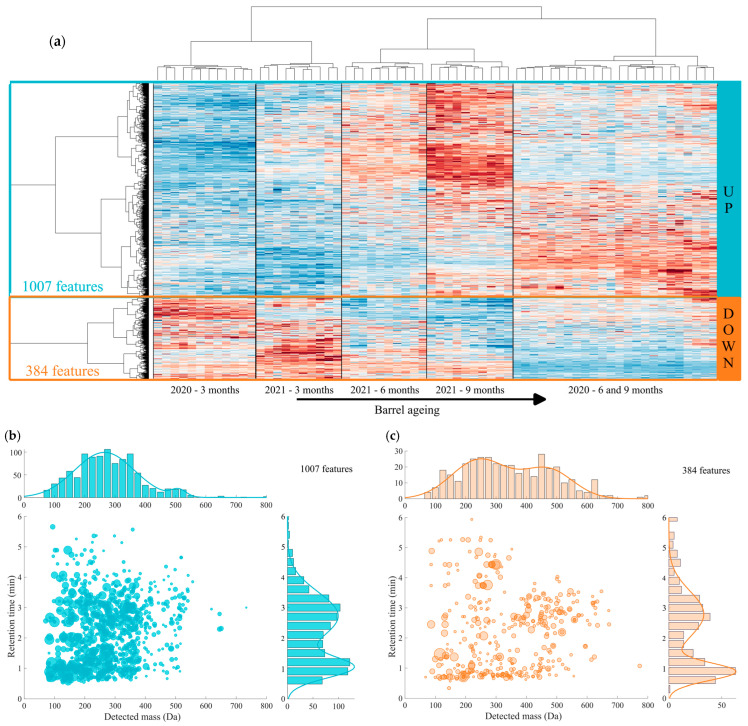
(**a**) Heatmap based on 1391 features significantly impacted during barrel ageing for CH-C-BWs from 2020 and 2021 vintages with barrel ageing in new oak barrels, showing the increasing (UP = 1007 features) and decreasing (DOWN = 384 features) features. Distribution of increasing (UP) (**b**) and decreasing (**c**) compounds according to their detected mass (Da) and retention time (min). The bubble size corresponds to the average intensity in all samples. Gaussian distributions have been calculated using $z_{1}=a_{1}\times \exp\left(-\frac{{\left(x-b_{1}\right)}^{2}}{c²}\right)$ for *m*/*z* and $z_{2}=a_{2}\times \exp\left(-\frac{{\left(y-b_{2}\right)}^{2}}{c_{2}²}\right)$ for retention time.

**Figure 5 antioxidants-13-00364-f005:**
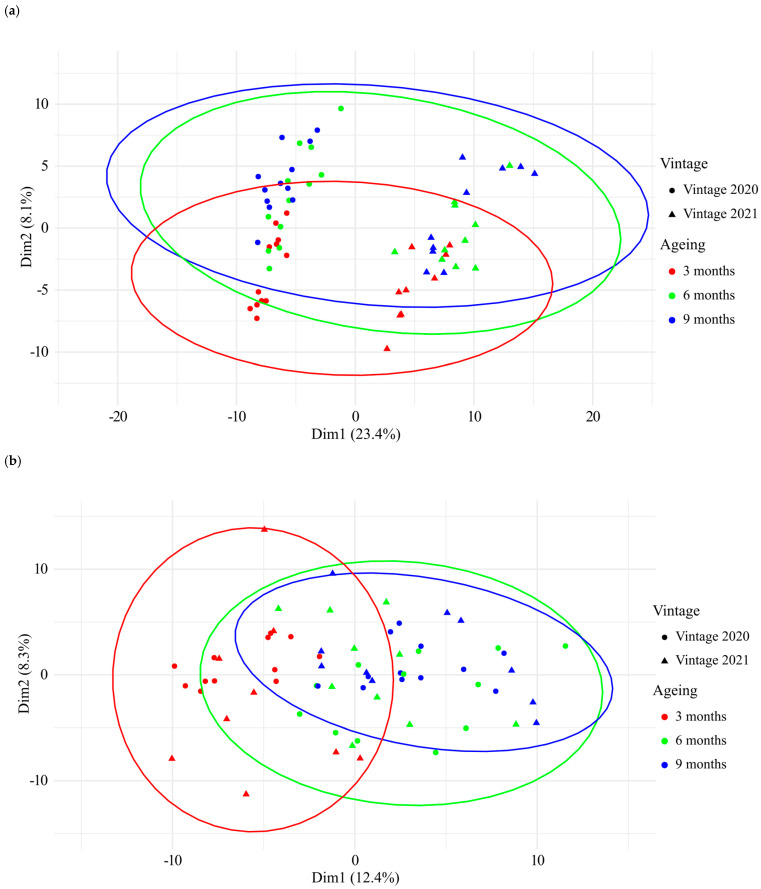
(**a**) Principal component analysis of UHPLC-Q-ToF-MS-based data of 235 AM-NU and their intensities observed in 66 CH-C-BWs from vintages 2020 and 2021 aged in new oak barrels, at 3, 6 and 9 months. (**b**) PCA after Pareto scaling by vintage. For the analysis of AM-NU, one sample was removed as an outlier (sample T3-Allier barrel 2 from vintage 2021).

**Figure 6 antioxidants-13-00364-f006:**
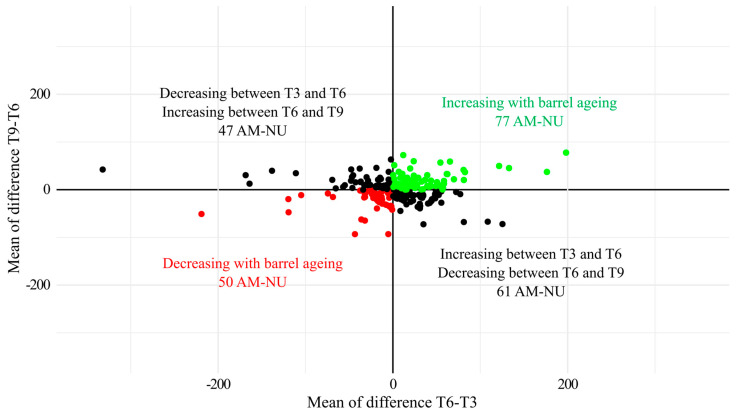
Dynamics of AM-NU after Pareto scaling by barrel ageing. For the analysis of AM-NU, one sample was removed as an outlier (sample T3-Allier barrel 2 from vintage 2021).

## Data Availability

Data is contained within the article and Appendix A.

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
