# Peer review of "The Oxidative Stability of Champagne Base Wines Aged on Lees in Barrels: A 2-Year Study"

_antioxidants, 2024, doi:10.3390/antiox13030364_

Round 1
Reviewer 1 Report
The manuscript is well-written, very interesting, and rich in information regarding the antioxidant capacity of CH-C-BW during fermentation and aging in oak barrels. However, some changes, particularly in result interpretation and language, could enhance the clarity of the manuscript and make it more accessible to a wider audience.
The manuscript by Charlotte Maxe et al. examines the evolution of antioxidant compound profiles and antioxidant activity of Chardonnay Champagne base wines (CH-C-BW) during fermentation and aging in new oak barrels. The authors consider two consecutive vintages, 2020 and 2021, for their study. The antioxidant activity of the samples is assessed using a solution assay (DPPH), while both targeted (ellagitannin concentrations) and untargeted (chemical fingerprints) analytical approaches are employed to evaluate changes in the metabolomic profile during the aging period. The results demonstrate that aging CH-C-BW in new oak barrels improves or maintains their antioxidant stability, also as a consequence of ellagitannin extraction from the oak. Specifically, despite the differences observed in initial antioxidant capacity between the 2020 and 2021 vintages, the consistent extraction profile in both vintages suggests uniform tannic potentials among the tested oak barrels.
The manuscript is well-written, very interesting, and rich in information regarding the antioxidant capacity of CH-C-BW during fermentation and aging in oak barrels. However, some changes, particularly in result interpretation and language, could enhance the clarity of the manuscript and make it more accessible to a wider audience. Specifically, it would be beneficial to better emphasize the implications of the findings. For example, what consequences could the observed changes in antioxidant metabolite profiles during aging have on the overall quality or aging potential of the wines? Some sections of the paper are dense with technical terminology, which may make it challenging for non-experts in the field to read. Simplifying the language where possible and providing clear explanations for specialized terms would improve accessibility and comprehension.
Author Response
Dear Reviewer,
First, we would like to thank you for your time and your useful comments. You have contributed to the improvement of this paper and to the reconsiderations of the final conclusions. Taken your suggestions into account, we added one sentence on the consequences of the changes in antioxidant metabolites concerning the ageing potential of wines as well as the quality of wines, especially in terms of perceptual properties (sweetness, bitterness and acidity).
To make clearer explanations, some precisions have been added.
We hope these precisions satisfy your expectations.
Regards
Reviewer 2 Report
The manuscript ID: antioxidants-2897965 entitled “ The oxidative stability of champagne base wines aged on lees in barrels: a 2-year study” by Maxe et al. can be published in Antioxidants after minor revision.
Specific comments
According to the Authors, the oxidative stability and related molecular fingerprints of Chardonnay champagne base wines (CH-C-BW) were tested after 1-year of on-lees ageing in new oak barrels for two consecutive vintages. Therefore, targeted argeted analyses were made for the determining CH-C-BW antioxidant capacities, and ellagitannins concentrations. Moreover, untargeted analyses were also performed for screening CH-C-BW chemical fingerprints. Next, statistical and chemometric evaluation of the obtained data was carried out. The study showed that for both vintages on-lees ageing in new oak barrels provided to improve or maintain their antioxidant stability determined by the DPPH assay at 1-year, while the extraction of total ellagitannins showed a linear profile. UHPLC-Q-ToF-MS molecular profiling based on 66 CH-C-BW confirmed a chemical change during the barrel ageing with 1007 features that significantly increased with the barrel ageing and 384 features that significantly decreased with the barrel ageing. Moreover, 235 features contributed to the molecular antioxidant markers with nucleophilic properties (AM-NU). Although the population of AM-NU molecular fraction was stable during barrel ageing, however, individual nucleophiles had stable, increasing or decreasing trendency according to the chemical environment related to the wine matrix complexity and the barrel ageing conditions. It confirms that vines antioxidant metabolome composed by antiradical and nucleophilic compounds clearly appeared vintage and barrel aging dependent. Thus, the results of the study allow to improve the knowledge of white wines antioxidant metabolome and the ageing potential of Chardonnay champagne base wines by integrating vintage and barrel ageing effects. The topic of the manuscript can be interesting for the readers, and this paper can be published in Antioxidants after minor revision.
3.3. Chardonnay base wines antioxidant metabolome during barrel ageing
The authors reported that PCA based on the 235 isolated features clearly revealed an impact of the vintage (PC1), and an impact of the aging duration (PC2) (Figure 5a). Moreover, PCA based on normalized dataset describing the intensity of each AM-NU by Pareto scaling in each sub-dataset confirmed a clear impact of the aging duration (Figure 5b). In my opinion, clear differences in respect to ageing can be observed for 3-moths old vines while these differences were as not significant between 6- and 9-moth vines Additionally, the explained variability of data sets by PC1 and PC2 in Figures 5 a and b was relatively low (31.5 and 20.7%, respectively). It should be commented.
2. In the manuscript there are editorial mistakes which should be corrected e.g.:
1. 2.7 Antioxidant metabolome and the whole manuscript. The reference number of the publication: Romanet et al., 2023; Romanet et al., 2020; Nikolantonaki & Waterhouse, 2012 Schymanski et al., 2014 (supplementary material) should be reported.
Line 34: lees [2] .
Line 108: (1mol/L
Lines 263 and 307: (n=10),
Lines 288 and 306: n=12
Line 434: [53,54] Indeed
Supplementary material: The title of Table S3 should be described above this table.
Summarizing, the manuscript can be published in Anitoxidants after minor revision.
Author Response
Dear Reviewer,
First, we would like to thank you for your time and your useful comments. You have contributed to the improvement of this paper and to the reconsiderations of the final conclusions. Concerning the low explained variability of PC1 and PC2 in Figures 5a and b, it is important to consider that the data set (66 white wines) is very different. These differences are based on the vintage and also on the chemical composition. Indeed, Chardonnay champagne base wines come from vintage 2020 have performed the MLF and not the ones from vintage 2021. The low explained variabilities of PC1 and PC2 are also justified by the fact that the two dimensions tried to explain the impact of the 235 variables (AM-NU), which is a huge number, compared to the number of wine samples.
we would especially thank you for detailing all the editorial mistakes. we added all the reference number of publications and corrected all the mistakes you highlighted.
We also checked the title of the Table S3 and put it above the Table S3.
we hope these precisions satisfy your expectations.
Regards
Reviewer 3 Report
An interesting work. It is the first report about the oxidative stability of champagne base wines aged in new barrels with lees. The results will give the winemakers of sparkling wine more knowledge about the effect of oak on base wine.
1.Line 63-64. The authors pointed out that " base wines 63 exhibit specific features compared to a still wine from the same grape variety, including 64 in particular lower pHs and higher titratable acidit", why not have a still wine as cotrol?2.Line 113. Can the authors explain why only choose DPPH method to analyze the antioxidant capacity. 3.Figure1. Can the authors explain why there is big difference of EC20 between 2020 and 2021 vintage? why EC20 of 2021 vintage have such a trend with aging.
Author Response
Dear Reviewer,
First, we would like to thank you for your time and your useful comments. You have contributed to the improvement of this paper and to the reconsiderations of the final conclusions.
For your comment line 63-64, champagne base wines are still wines used for the champagne-making process but present specific properties (lower pH and higher titratable) to other white still wines such as Chardonnay from Bungundy. No control (base wines aged in stainless-steel tanks) has been used for this study.
The DPPH method to analyze the antioxidant capacity was used for different reasons. The DPPH assay is simple, inexpensive and an efficient method to determine the antioxidant capacity of biological matrices [1–4]. Moreover, the radical DPPH can be reduced by amino acids containing cysteine and aromatic amines [5]. The adapted DPPH used in this article is specially appropriated to white wines and allowed us to access to the sulfur-containing compounds that contribute to the antioxidant capacity of white wines [6]. Wine antioxidant metabolome was also studied using untargeted analyses leading to the establishment of the AM-NU and AM-RS fractions, as defined by a previous study [7].
The big difference of Ec20 of vintages 2020 and 2021 can be explained by the vintage and the climatic conditions, that are known to be one of the major parameters to explain wine antioxidant capacity [8]. There are also vineyard conditions that contribute too, but it is important to have in mind that grapes come from the same field for both vintages, so growing locations and rootstocks and cultivars do not explain this variability [8].
we would especially thank you for your interest in our study. We added two sentences to clarify the potential reason that could explain the big difference of Ec20 between both vintages.
we hope these precisions satisfies your expectations.
Regards
- Carmona-Jiménez, Y.; García-Moreno, M. de V.; Igartuburu, J.M.; Garcia Barroso, C. Simplification of the DPPH Assay for Estimating the Antioxidant Activity of Wine and Wine By-Products. Food Chem. 2014, 165, 198–204, doi:10.1016/j.foodchem.2014.05.106.
- Fernández-Pachón, M.S.; Villaño, D.; García-Parrilla, M.C.; Troncoso, A.M. Antioxidant Activity of Wines and Relation with Their Polyphenolic Composition. Anal. Chim. Acta 2004, 513, 113–118, doi:10.1016/j.aca.2004.02.028.
- Popovici, C.; Saykova, I.; Tylkowski, B. Evaluation de l’activité Antioxydant Des Composés Phénoliques Par La Réactivité Avec Le Radical Libre DPPH. Rev. Génie Ind. 2009, 4, 25–39.
- Comuzzo, P.; Battistutta, F.; Vendrame, M.; Páez, M.S.; Luisi, G.; Zironi, R. Antioxidant Properties of Different Products and Additives in White Wine. Food Chem. 2015, 168, 107–114, doi:10.1016/j.foodchem.2014.07.028.
- Blois, M.S. Antioxidant Determinations by the Use of a Stable Free Radical. Nature 1958, 1199–1200, doi:10.1038/1811199a0.
- Romanet, R.; Coelho, C.; Liu, Y.; Bahut, F.; Ballester, J.; Nikolantonaki, M.; Gougeon, R.D. The Antioxidant Potential of White Wines Relies on the Chemistry of Sulfur-Containing Compounds: An Optimized DPPH Assay. Molecules 2019, 24, 1353, doi:10.3390/molecules24071353.
- Romanet, R.; Gougeon, R.D.; Nikolantonaki, M. White Wine Antioxidant Metabolome: Definition and Dynamic Behavior during Aging on Lees in Oak Barrels. Antioxidants 2023, 12, 395, doi:10.3390/antiox12020395.
- Lachman, J.; Šulc, M.; Faitová, K.; Pivec, V. Major Factors Influencing Antioxidant Contents and Antioxidant Activity in Grapes and Wines. Int. J. Wine Res. 2009, 1, 101–121, doi:10.2147/IJWR.S4600.